# Fatty Acid Analysis, Chemical Constituents, Biological Activity and Pesticide Residues Screening in Jordanian Propolis

**DOI:** 10.3390/molecules26165076

**Published:** 2021-08-21

**Authors:** Rajashri R. Naik, Ashok K. Shakya, Ghaleb A. Oriquat, Shankar Katekhaye, Anant Paradkar, Hugo Fearnley, James Fearnley

**Affiliations:** 1Faculty of Pharmacy, Al-Ahliyya Amman University, Amman 19328, Jordan; rsharry@ammanu.edu.jo; 2Pharmacological & Diagnostic Research Center, Faculty of Pharmacy, Al-Ahliyya Amman University, Amman 19328, Jordan; 3Faculty of Allied Medical Sciences, Al-Ahliyya Amman University, Amman 19328, Jordan; goreqat@ammanu.edu.jo; 4Natures Laboratory Ltd., Whitby YO22 4NH, UK; s.katekhaye@bradford.ac.uk (S.K.); hugo.fearnley@beevitalpropolis.com (H.F.); 5Centre for Pharmaceutical Engineering Science, School of Pharmacy, University of Bradford, Bradford BD7 1DP, UK; a.paradkar1@bradford.ac.uk; 6Apiceutical Research Centre, 3b Enterprise Way, Whitby YO22 4NH, UK; james.fearnley@beevitalpropolis.com

**Keywords:** propolis, GC-FID, LC-MS-MS, HPLC-PDA, desmedipham, pinocembrin, chrysin, caffeic acid phenyl ester, antioxidant, xanthine oxidase inhibition

## Abstract

Propolis is a resinous natural product collected by honeybees (*Apis mellifera* and others) from tree exudates that has been widely used in folk medicine. The present study was carried out to investigate the fatty acid composition, chemical constituents, antioxidant, and xanthine oxidase (XO) inhibitory activity of Jordanian propolis, collected from Al-Ghour, Jordan. The hexane extract of Jordanian propolis contained different fatty acids, which are reported for the first time by using GC-FID. The HPLC was carried out to identify important chemical constituents such as fatty acids, polyphenols and α-tocopherol. The antioxidant and xanthine oxidase inhibitory activities were also monitored. The major fatty acid identified were palmitic acid (44.6%), oleic acid (18:1∆^9^*cis*, 24.6%), arachidic acid (7.4%), stearic acid (5.4%), linoleic acid (18:2∆^9–12^*cis*, 3.1%), caprylic acid (2.9%), lignoceric acid (2.6%), *cis*-11,14-eicosaldienoic acid (20:2∆^11–14^*cis*, 2.4%), palmitoleic acid (1.5%), *cis*-11-eicosenoic acid (1.2%), α–linolenic acid (18:3∆^9–12–15^*cis*, 1.1%), *cis*-13,16-docosadienoic acid (22:2∆^13–16^*cis*, 1.0%), along with other fatty acids. The major chemical constituents identified using gradient HPLC-PDA analysis were pinocembrin (2.82%), chrysin (1.83%), luteolin-7-*O*-glucoside (1.23%), caffeic acid (1.12%), caffeic acid phenethyl ester (CAPE, 0.79%), apigenin (0.54%), galangin (0.46%), and luteolin (0.30%); while the minor constituents were hesperidin, quercetin, rutin, and vanillic acid. The percentage of α-tocopherol was 2.01 µg/g of the lipid fraction of propolis. Antioxidant properties of the extracts were determined via DPPH radical scavenging. The DPPH radical scavenging activities (IC_50_) of different extracts ranged from 6.13 to 60.5 µg/mL compared to ascorbic acid (1.21 µg/mL). The xanthine oxidase inhibition (IC_50_) ranged from 75.11 to 250.74 µg/mL compared to allopurinol (0.38 µg/mL). The results indicate that the various flavonoids, phenolic compounds, α-tocopherol, and other constituents which are present in propolis are responsible for the antioxidant and xanthine oxidation inhibition activity. To evaluate the safety studies of propolis, the pesticide residues were also monitored by LC-MS-MS 4500 Q-Trap. Trace amounts of pesticide residue (ng/mL) were detected in the samples, which are far below the permissible limit as per international guidelines.

## 1. Introduction

Propolis is a resinous natural product collected by honeybees (such as *Apis mellifera* and others) from tree exudates or resins with bee-wax along with salivary secretions used in natural medicine for long time. Bees use the propolis to repair the cracks and strengthen the walls of the hives to create a narrow and defendable entry into the hive. Bees also use propolis to polish the interior of their hives to control the proliferation of pathogenic microorganisms. Propolis also helps to remove the entry of intruders and to fix their corpses [1].

Many pharmacological activities of propolis are well documented, such as free radical scavenging, antioxidant, anticancer, anti-bacterial, anti-inflammatory, hepato-protective, radio-protective and neuro-protective [2,3,4,5,6]. The pharmacological and biological effects of propolis depend on their geographical location, altitude, time of collection, season and the species of the bee. The chemical composition of propolis from differing geographic regions is well recorded in the literature [7,8,9,10,11,12]. There are major chemical differences between propolis from China, India, Chile, South Africa, Taiwan and other countries [7,8,9,10]. Due to above mentioned factors, the chemical composition of the propolis is highly variable. The color, flavor and texture also vary from season to season. Due to the wide diversity of propolis, it is always in demand to explore for its pharmacological and biological activity [10,11,12].

In general, the propolis contains flavonoids and terpenes, as these chemical constituents are present in the flowers and other parts of the plant. Cuban propolis contains polyisoprenylated benzophenones, while the Brazilian propolis contains derivatives of acetophenone and p-coumaric acid. Chilean propolis contains benzopyran, phenylpropane and other aromatic aldehydes and derivatives [13].

In pre-Columbian culture the consumption of honey and culture of *Melipona beecheii* (bee without sting) was very common. During the 15th century the beekeeping was developed with the European bee (*Apis mellifera*). Jordanian propolis contains chemical compounds such as 4(Z)-1-3-dihydroxyeupha-7,24-dien-26-oic acid [2], along with other compounds such as pinobanksin-3-*O*-acetate, pinocembrin and chrysin [3] and lignoceric acid [2]. It has been reported that other propolis contains chrysin and kaempferol, which are used as anti-aging agents due to their antioxidant activity [14,15,16,17,18,19]. Since ancient times, propolis has been used for its therapeutic potentials in various ailments. It still exists in some of the traditional medicines in Eastern Europe and some parts of the world. Based on this, studies are ongoing in many parts of the world on the biological properties of propolis, due to which it has been used in various therapeutical applications, such as ointments and creams to heal wounds, treat burns and skin problems and ulcers. Various forms of propolis preparations have also been used in the treatment of laryngological problems, gynaecological diseases, asthma and diabetes, and it is also used in toothpaste and mouthwash preparations to combat gingivitis and stomatitis. There is a considerable amount of reported literature on the use of propolis in cosmetics for lotions, face creams and solutions [20,21,22].

Literature surveys have shown that the fatty acid composition and biological activity of Jordanian propolis has not been explored in great detail, hence it was thought worthwhile to study the composition of Jordanian propolis and explore its antioxidant and xanthine oxidase inhibitor activity.

## 2. Results and Discussion

### 2.1. Determination of Fatty Acid Methyl Esters (FAME) by GC-FID

As mentioned earlier, propolis is a resinous nontoxic product of the hive, primarily resin collected by honeybees from trees and plants. Propolis is known to have several health benefits for human beings and has been used since time immemorial [23,24]. The pharmacological and the chemical composition of propolis may vary significantly depending on the geographical location, time and seasonal variations [7,25]. The present study was carried out to investigate the fatty acid composition, antioxidant and xanthine oxidase inhibitory activity of Jordanian propolis. Propolis was collected from the Al-Ghour region of the Jordan. The analysis was carried out by GC-FID and 19 compounds were identified. Most of the compounds that were identified are presented in Table 1. Different fatty acid methyl esters of the propolis were identified using standard FAME which contains 37 methyl esters of C_4_–C_24_ fatty acid. The hexane extract of Jordanian Propolis contains different fatty acids, some of which are reported for the first time. The major fatty acid identified were palmitic acid (44.5%), oleic acid (18:1∆^9^*cis*, 24.6%), arachidic acid (7.4%), stearic acid (5.4%), linoleic acid (18:2∆^9–12^*cis*, 3.1%), caprylic acid (2.9%), lignoceric acid (2.6%), *cis*-11,14-eicosadienoic acid (20:2∆^11–14^-*cis*, 2.4%), palmitoleic acid (1.5%), *cis*-11-eicosenoic acid (1.2%), α–linolenic acid (18:3∆^9–12–15^-*cis* 1.1%), *cis*-13,16-docosadienoic acid (22:2∆^13–16^-*cis*, 1.0%), along with other fatty acids (Figure 1). Fatty acids such as oleic (18:1), palmitic (16:0), linoleic (18:2), and stearic (18:0), were reported by Castro et al. and Duarte et al. [26,27] in propolis samples. A total of 10 compounds were identified by Thirugnanasampandan et al. [28] from the propolis collected from the Tamilnadu region of India, using GC-MS to show the presence of fatty acids. Among 10 compounds the major fatty acid present were as 9-octadecenoic acid (3.2%), decanoic acid (2.12%) 9,12-hexadecadienoic acid (1.29%), octadecadienoic acid methyl ester (0.49%) and alcohols such as 1-tetradecanol (0.89%), octadecanol (0.69%), 1-dotricontanol (0.48%) and 2,3-epoxy-5,8-hectadecadien-1-ol (0.6%) [28]. Fourteen fatty acid methyl esters were identified by GC-MS method in propolis by Sahinler and Kaftanoglu [29]. FAME identified were hexadecanoic acid, 9-octadecanoic acid, docosanoic acid, tetracosanoic acid, hexacosanoic acid, octacosanoic acid, triacontanoic acid, octadecanoic acid, 8-octadecanoic acid, 9,12,15-octadectrienoic acid, hexacosanoic acid, 9,12-octadecanoic acid and pentanoic acid methyl esters. Hegazi and El Hady [30] analyzed the chemical constituents of propolis from different province of Egypt using GC-MS, they reported 71 compounds of which 14 were new to propolis. In other study, they have reported 75 compounds of which 22 were new to propolis [31]. In both the studies the percent composition of the chemical constituents varied from different region of the province. Around 50 individual compounds were identified by Popova et al. [32] in the propolis collected from Oman. These compounds include sugars, polyols, hydroxy acids, fatty acids, cardanols and cardols, anacardic acids, flavan derivatives, triterpenes, prenylated flavanones and chalcones. They observed that not all propolis were similar in their chemical profiling. Most of the available reports are based on the GC-MS. We have identified for the first time 19 compounds by GC-FID. Composition and concentration showed significant disparity between the types of propolis which may be attributed to the geographical location, time, and seasonal variation. Bankova et al. [33] demonstrated that the plant source of the propolis differs significantly even in the same beehive and in the same season. Due to this disparity in the chemical constituents there may be difference in the biological activity such as antioxidant and xanthine oxidase activity.

### 2.2. Analysis of Chemical Constituents Using HPLC-PDA

The major chemical constituents identified using gradient HPLC-PDA analysis were pinocembrin (2.82%), chrysin (1.83%), luteolin-7-*O*-glucoside (1.23%), caffeic acid (1.12%), caffeic acid phenethyl ester (CAPE, 0.79%), apigenin (0.54%, galangin (0.46%), luteolin (0.30%) and minor constituents such as hesperidin, quercetin, rutin, vanillic acid (Table 2 and Figure 2). The percentage of α-tocopherol was 2.01 µg/g of lipid fraction of propolis. Romero et al. [34] identified 21 compounds in Brazilian propolis. The identified compounds were (1) caffeic acid, (2) p-coumaric acid, (3) ferulic acid, (4) 3,4-dimethyl-caffeic acid, (5) pinobanksin-5-methyl-ether, (6) kaempferide, (7) apigenin, (8) kaempferol, (9) cinnamilidenacetic acid, (10) caffeic acid prenyl ester, (11) chrysin, (12) pinocembrin, (13) galangin, (14) caffeic acid phenylethyl ester, (15) pinobanksin-3-*O*-acetate, (16) p-coumaric prenyl ester, (17) p-coumaric cinnamyl ester, (18) pinobanksin-3-*O*-butyrate, (19) pinobanksin-3-*O*-pentanoate, (20) pinobanksin-3-*O*-hexanoate, and (21) p-methoxy cinnamic acid cinnamyl ester.

### 2.3. LC-MS-MS Screening for Pesticide Residues in Propolis

The propolis was screened for more than 400 pesticides using the QuEChERS method according to ABSciEx guidelines. All the targeted analyses were carried out using the LC-MS-MS-4500-QTrap. The QuEChERS method (Quick-Easy-Cheap-Effective-Rugged-Safe extraction) has been developed for the determination of pesticide residues using earlier reported methods [35]. This is the first report on Jordanian propolis which shows the presence of trace amount of pesticide residues (in ng/mL concentration) in natural product. The propolis contains (trace amount of) desmedipham (37.41 ± 0.70), fenpropomorph (21.52 ± 0.51), dichlfenthion (15.83 ± 0.33) and etoxazole (20.94 ± 0.25) in ng/mL concentration which might be due to the use of the pesticide in the country side (Table 3 and Figure 3). The amount of pesticide residues in the propolis is within the acceptable range as per guideline [36,37,38,39].

### 2.4. Biological Activity

#### 2.4.1. DPPH Radical Scavenging Activity

One of the important characteristics of propolis is its antioxidant and antiradical activity [9]. Antioxidant properties of the different extracts were determined via DPPH radical scavenging, β-carotene bleaching assay and NO scavenging assay. The extract produced significant antioxidant activity in vitro with free radical scavenging activity with IC_50_ value of 6.13 ± 0.1, 14.4 ± 0.1 and 60.5 ± 0.1 µg/mL of the 70% ethanolic, 50% ethanolic and lipid extraction (Table 4). The IC_50_ values for ascorbic acid (in 50% ethanol) and α-Tocopherol (in hexane) were 1.21± 0.03 and 85.5 ± 1.7 µg/mL respectively. Similar results were reported by Bankova et al. (2019) [33] in the Romanian propolis with significant scavenging activity positively correlated to the presence of flavonoids. Similar observations were made by da Silva et al. (2006) [40] and Ahn et al. (2007) [41] on Brazilian and Chinese propolis respectively. Similar results were reported by Thirugnanasampandan et al. on the antioxidant activity of Indian propolis [28]. It may be noted that the propolis from different geographical locations exhibit antioxidant activity despite the disparity or difference in their chemical composition [33,42]. The antioxidant activity in the propolis may be attributed to the presence of major compounds such as flavonoids and phenolic acids. These compounds are useful as natural antioxidants and prevent oxidative damage of DNA caused by ROS [43]. The antioxidant effect may be due to the scavenging activity of the free radical and the interaction with enzymes. As reported by Moreira et al. (2008) [44] some of the chemical components of the propolis are absorbed and circulated in the blood and these compounds act as hydrophilic antioxidants and save vitamin C. Several bee products also exhibit the antioxidant activity which is attributed to the presence of flavonoids [45].

#### 2.4.2. In Vitro Xanthine Oxidase Inhibition Activity

There is not much literature available on the inhibitory activity of propolis on xanthine oxidase and treatment of gout. The propolis extracts exhibited appreciable xanthine oxidase inhibitory activity in vitro. Results of the xanthine oxidase (XO) activity is presented in the Table 4 and the results are expressed as inhibitory concentration (µg/mL). It was observed that the extracts inhibited the xanthine oxidase with IC_50_ value of 75.11 ± 11.43, 89.51 ± 0.17.40 and 250.74 ± 13.09 µg/mL of the 70% ethanolic, 50% ethanolic and lipid fraction respectively. The lower IC_50_ value in 70% ethanolic extract might be due to the presence of hydrophilic as well as lipophilic compounds. The value of standard compound allopurinol against xanthine oxidase was 0.36 ± 0.08 µg/mL. Similar inhibitory activity was reported by Russo et al. [46] due to the presence of CAPE and galangin. In eukaryotic cell xanthine oxidase is an enzyme that is the source of superoxide anions. Most of the natural compound such as some polyphenols exhibit a dose dependent inhibition of xanthine oxidase [47]. One of the characteristic of the ischemic injury is the over production of superoxide anion due to the leak of electron in the mitochondrial respiratory chain and due to the generation by the conversion of xanthine dehydrogenase to xanthine oxidase that produce superoxide anion when it oxidizes xanthine into uric acid [48]. Flavonoids and phenolic acids may be one of the potent inhibitor against the metabolic enzymes such as cyclo-oxygenase, xanthine oxidase and lipo-oxygenase [49], which can control the diseases such as inflammation, hyperuricemia and gout. Hence compounds such as flavonoids and flavones glycoside such as rutin, luteolin, luteolin-7-O-glucoside, quercetin, apigenin, pinocembrin, chrysin, CAPE, galangin and hesperidin, may have a role in the inhibition of XO. XO inhibitor and uricosuric agents are used in the treatment of diseases such as gouty arthritis and inflammatory disease. The drug allopurinol is used in the treatment of gout, but these drugs are associated with minor side effects [50], hence drug with lesser side effect and more therapeutic activity is required.

## 3. Materials and Methods

The DPPH^•^, 4-hydroxy coumaric acid, apigenin, apigenin-7-*O*-glucoside, caffeic acid, CAPE, carnosic acid (used as IS), chlorogenic acid, chrysin, galangin, gallic acid, hesperidin, lueolin-7-*O*-glucoside, luteolin, naringenin, pinocembrin, quercetin, rosmarinic acid, rutin, vanillic acid, xanthine oxidase (Bovine), allopurinol, xanthine, were procured from Sigma-Aldrich (St. Louis, MO, USA). Thirty-seven methyl esters of C_4_–C_24_ fatty acids, ascorbic acid, α-tocopherol of analytical grade were purchased from Sigma-Aldrich Chem. Co. (St. Louis, MO, USA).

### 3.1. Collection of Samples

Jordanian propolis was purchased from a local market. The propolis was originally collected from beehives located at Al-Ghour region in Jordan from March to July 2018 using propolis collectors. Propolis from the collectors was gathered and kept at 20 °C until processed. The ground propolis (10 g) was extracted with 100 mL of ethanol (either 70% or 50%) or with hexane at 25 °C for 48 h (*n* = 3). The extracts were then filtered through a Whatman no. 1 filter paper and the solvent was evaporated using Buchi R-100 Rotary Evaporator (BÜCHI Labortechnik AG, Flawil, Switzerland). The samples were stored under nitrogen till use.

### 3.2. Analysis of Samples

The extracts were concentrated by vacuum evaporation, reconstituted and then filtered by syringe filter with a 0.22 mm membrane. The extracts were then tested for chemical constituents using GC-FID, and high-performance liquid chromatography (HPLC). The α-tocopherol content was determined in the lipid fraction (hexane extract). For the monitoring of pesticide residues, a crude sample of propolis was used.

#### 3.2.1. Determination of Fatty Acid Methyl Esters (FAME) by GC-FID

FAME was synthesized by using sodium methoxide in presence of methanol at 40 °C. In brief, a solution of fixed oil (0.1 g) in methanol (25 mL), sodium methoxide solution (30% *w*/*v* in methanol, 0.15 g) was added with stirring. The reaction mixture was maintained at 40 °C for 45 min with constant shaking using Memmert water bath and shaker (Memmert GmbH Co. KG, Schwabach, Germany). Twenty-five milliliters of *n*-hexane was added, and the solution was shaken for 20 min. The reaction was stopped using saturated solution of oxalic acid. The precipitated sodium oxalate was removed after centrifuging the mixture at 5000× *g* rpm for fifteen minutes. The supernatant was collected and dried over anhydrous sodium sulphate and were analyzed by GC-FID for FAME.

FAME samples were analyzed using gas chromatograph (Model 2030, Nexera, Shimadzu, Japan) [51]. The instrument is equipped with DB-23 capillary column with thickness of the film of 0.25 μm, length of 60 m, and 0.250 mm internal diameter. The optimum conditions for operating the GC with respect to temperature were as follows: initial temperature of the oven was maintained at 50 °C for 1 min and then raised to 175 °C (@25 °C/min) then it was raised to 230 (@4 °C/min), there after the temperature was maintained at 230 °C for 10 min, separation of analytic were achieved by carrier gas (He) at a Linear velocity of 33 cm/s. The injection volume of samples was 1 µL and the split ratio was 1:50. The injector port and the detectors were maintained at a temperature of 280 °C with a total run time of 40 min. The signals were recorded using Windows 7 based GC solution software (Version 1.25, Shimadzu Corporation, Kyoto, Japan) and the data were analyzed. Different fatty acid methyl esters of the propolis were identified, using standard FAME which contains 37 methyl esters of C_4_–C_24_ fatty acid (Sigma-Aldrich Chem. Co., St. Louis, MO, USA). The results of the three independent reactions were averaged on the basis of three different experiments.

#### 3.2.2. HPLC-PDA Analysis of Propolis

##### Determination of Chemical Constituents

Phenolic compounds in the samples of propolis (70% ethanolic extract) were identified and quantified using a Nexera-2030-3D integrated HPLC instrument (Shimadzu, Kyoto, Japan) equipped with a quaternary pump and a PDA detector, sample cooler and column oven. Quantitative and qualitative analyses were carried on Hypersil—Gold C18 column (4.6 mm i.d. × 250 mm, 5.0 μm particle size, Thermo-Fisher, Waltham, MA, USA). Chromatograms were acquired between 190 to 400 nm and processed using the tools of the Lab Solution software (Shimadzu Corporation, Kyoto, Japan). The flow rate used for elution was 1 mL/min and signals were monitored by UV detection at 280 nm. The sample injection volume was 10 μL. The solvent system was 2% (*v*/*v*) glacial acetic acid in water (solvent A) and acetonitrile (solvent B). The gradient elution program was as follows: initially 95% A (*v*/*v*) and 5% B at 0–1 min, 95–75% A (*v*/*v*) at 1–15 min, 75–20% A (*v*/*v*) at 15–50 min, 20–0% A (*v*/*v*) at 50–60 min, 0% A (*v*/*v*) was maintained up to 63 min, thereafter 0–95% A (*v*/*v*) at 63–68 min and then 95% A (*v*/*v*) was maintained up to 74 min. Identification of the compounds in the chromatograms were performed by the comparison of their retention times, UV spectra and peak purity profile with those of reference standards. Determination of each phenolic compound was performed using the corresponding calibration curve. Extract samples were injected three times for HPLC analysis.

##### Determination of α-Tocopherol in Lipid Fraction

α-Tocopherol content in the lipid fraction was determined using the procedure described earlier [51]. Briefly, the separation of different isomers and quantization was carried out using BDS-Hypersil column (150 mm × 4.6 mm, 5 µm) using a mixture of methanol and acetonitrile (50:50, *v*/*v*) as mobile phase (flow rate—1 mL/min). The signals of analyte were captured using PDA detector and quantitated at 290 nm using LC-solution (Version 1.25, Shimadzu Corporation, Kyoto, Japan) software, after injecting 5 µL of different standard and test sample. The α-tocopherol content was calculated in the oil from the calibration curve.

#### 3.2.3. LC-MS-MS Screening for Pesticide Residues in Propolis

The propolis was screened for more than 400 pesticide residues using in-house developed QuEChERS method according to ABSciEx guideline using LC-MS-MS 4500-QTrap (ABSciex USA, AB Sciex LLC, Framingham, MA, USA). QuEChERS method has been developed for the determination of pesticide residues using earlier reported methods [35].

### 3.3. Biological Activity

#### 3.3.1. DPPH Radical Scavenging Activity

The samples of the propolis were analyzed for its free radical scavenging activity using DPPH radical according to the reported method with slight modification [52]. To perform the analysis, the solution of DPPH radical (0.008 g %) was prepared freshly in ethanol (95%) or normal hexane. Different concentrations of extracts or lipid fractions of samples (1000 µg/mL to 1.95 µg/mL) in methanol or hexane were prepared. The DPPH radical (1 mL) solution and the samples (1 mL) were mixed and vortexed for 45 s, and kept in dark at 25 ± 2 °C for around 25 min. The absorbance of the solutions was measured using Shimadzu UV-1800 spectrophotometer (Shimadzu, Kyoto, Japan) at 517 nm using hexane as blank. DPPH radical scavenging activity was determined and the IC_50_ was calculated as reported earlier.

#### 3.3.2. Xanthine Oxidase Inhibiting Activity

XO inhibitory activity was measured by monitoring uric acid formation in xanthine oxidase system as described previously [53]. The assay system consisted of 0.6 mL phosphate buffer (100 mM; pH 7.4), 0.1 mL sample, 0.1 mL XO (0.2 U/mL), and 0.2 mL xanthine (1 mM; dissolved in 0.1 N NaOH). The reaction was initiated by adding the enzyme with or without inhibitors. Changes in absorbance of the mixture at 290 nm for 15 min compared to the absorbance of reagent blank were determined. A 0.2 mL aliquot of 1 N HCl was used to stop the enzymatic reaction. Allopurinol was used as positive control.

### 3.4. Statistical Analysis

Results are expressed as mean ± standard deviation (SD). Graph-Pad Prism 5 (Graph-Pad Software, San Diego, CA, USA) for Windows was used for statistical analysis of experimental data.

## 4. Conclusions

The present study suggests that Jordanian propolis is rich in polyphenols, flavonoids and fatty acid derivatives which are responsible for their antioxidant and xanthine oxidase activity. The study also suggests that the pesticide contents were far below the permissible limit due to the controlled use of pesticides in the crops. Hence, the propolis can be exploited more for their therapeutical potential.

## Figures and Tables

**Figure 1 molecules-26-05076-f001:**
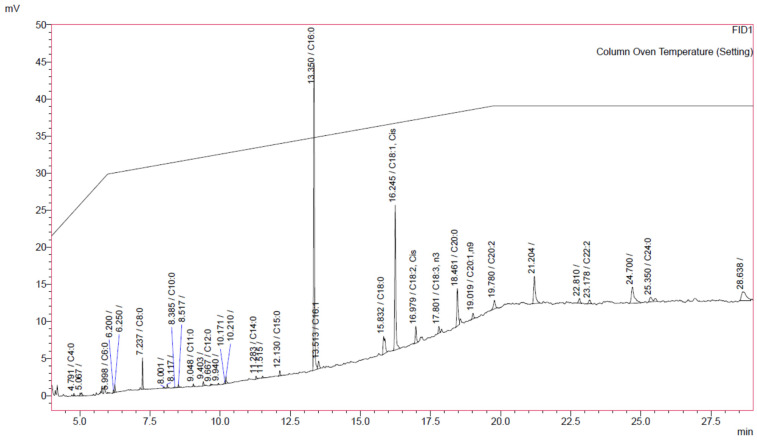
GC-FID analysis of Jordanian propolis sample, showing different identified FAME.

**Figure 2 molecules-26-05076-f002:**
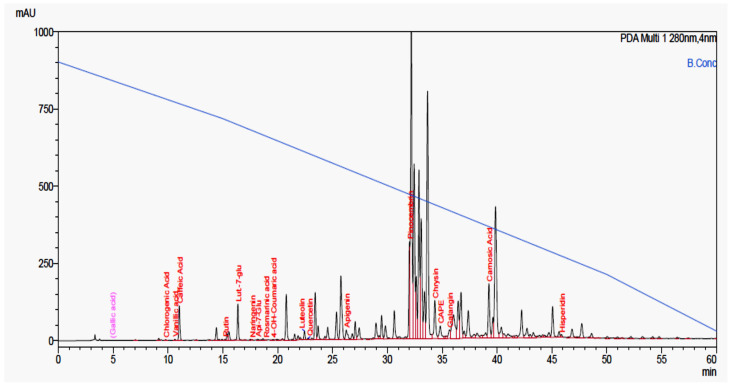
HPLC-PDA analysis of Jordanian propolis sample (70% ethanolic extract), showing different identified chemical constituents.

**Figure 3 molecules-26-05076-f003:**
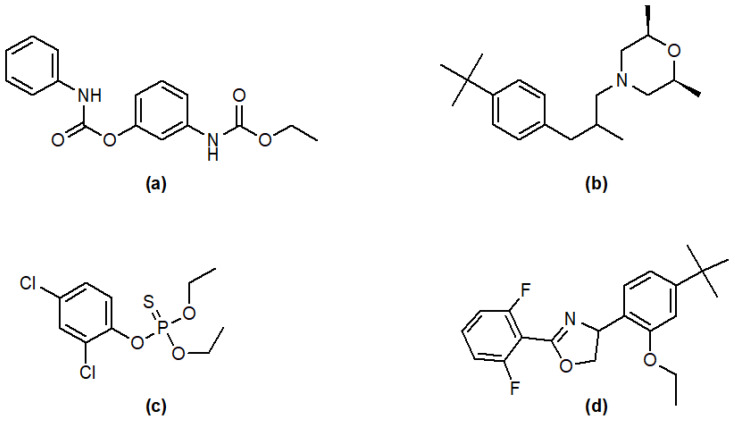
Chemical Structure of pesticides detected in the propolis sample, (**a**) desmedipham, (**b**) fenpropomorph, (**c**) dichlofenthion and (**d**) etoxazole.

**Table 1 molecules-26-05076-t001:** Fatty acid composition of propolis.

Time	Formula	Name (Identified as FAME)	Relative Percentage *
4.791	C4:0	Butyric acid	0.34
5.998	C6:0	Caproic acid	0.08
7.237	C8:0	Caprylic acid	2.93
8.385	C10:0	Capric acid	0.13
9.048	C11:0	Undecanoic acid	0.29
9.667	C12:0	Lauric acid	0.24
11.283	C14:0	Myristic acid	0.47
12.131	C15:0	Pentadecanoic acid	0.70
13.351	C16:0	Palmitic acid	44.55
15.832	C18:0	Stearic acid	5.42
18.461	C20:0	Arachidic acid	7.36
25.351	C24:0	Lignoceric acid	2.59
		**ƩSFA ^a^**	**65.10**
13.513	C16:1	Palmitoleic acid	1.52
16.245	C18:1, *cis*	Oleic acid	24.57
19.019	C20:1, n9	*cis*-11-Eicosenoic acid	1.15
		**ƩMUFA ^b^**	**27.24**
16.979	C18:2, *cis*	Linoleic acid	3.08
17.801	C18:3, n3	α-Linolenic acid	1.13
19.78	C20:2	*cis*-11,14-Eicosadienoic acid	2.40
23.178	C22:2	*cis*-13,16-Docosadienoic acid	1.05
		**ƩPUFA ^c^**	**7.66**

* Each value in the table represents the mean of three replicates; ^a^ SFA = saturated fatty acids; ^b^ MUFA = monounsaturated fatty acids; ^c^ PUFA = polyunsaturated fatty acids.

**Table 2 molecules-26-05076-t002:** Chemical constituents identified by HPLC-PDA in propolis sample collected from Jordan.

Time (min)	Name	Relative Percentage
4.900	Gallic acid	-
9.789	Chlorogenic acid	0.026
10.632	Vanillic acid	0.035
11.039	Caffeic acid	1.124
15.216	Rutin	0.036
16.372	Lueolin-7-*O*-glucoside	1.237
17.662	Naringenin	0.005
18.279	Apigenin-7-*O*-glucoside	0.037
18.917	Rosmarinic acid	0.002
19.583	4-hydroxy coumaric acid	0.026
22.446	Luteolin	0.301
22.774	Quercetin	0.063
26.258	Apigenin	0.540
32.023	Pinocembrin	2.819
34.328	Chrysin	1.828
340823	CAPE	0.790
35.659	Galangin	0.462
39.257	Carnosic acid (used as IS)	-
45.847	Hesperidin	0.103

**Table 3 molecules-26-05076-t003:** LC-MS-MS analysis of pesticide in the propolis.

Pesticide	Concentration (ng/mL) *
Desmedipham	37.41 ± 0.70
Fenpropomorph	21.52 ± 0.51
Dichlofenthion	15.83 ± 0.33
Etoxazole	20.94 ± 0.25

* *n* = 3.

**Table 4 molecules-26-05076-t004:** In vitro DPPH radical scavenging and XO activity of propolis.

Sample	IC_50_ (µg/mL)	
	DPPH Radical Activity *	XO Activity of Propolis *
Propolis (70% ethanolic extract)	6.13 ± 0.1	75.11 ± 11.43
Propolis (50% ethanolic extract)	14.4 ± 0.1	89.51 ± 17.40
Propolis (hexane extract, Lipid Fraction)	60.5 ± 0.1	250.74 ± 13.09
Ascorbic Acid (in 50% ethanol)	1.21 ± 0.03	-
α-Tocopherol (in hexane)	85.5 ± 1.7	-
Allopurinol	-	0.38 ± 0.08

* (*n* = 3).

## Data Availability

No new data were created or analyzed in this study. Data sharing is not applicable to this article.

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
