# Peer review of "Fatty Acid Analysis, Chemical Constituents, Biological Activity and Pesticide Residues Screening in Jordanian Propolis"

_molecules, 2021, doi:10.3390/molecules26165076_

Round 1
Reviewer 1 Report
All my comments are listed below with an appropriate Line number(s) from text in order to facilitate tracking:
- In the title of Manuscript it should be "pesticide residues" not just "pesticide".
Line 15: The given statement is not complet since not only honey bees collect propolis. Stingless bees do the same (https://doi.org/10.1016/j.phymed.2019.153098). Correct and fulfill.
Line 19: "contained" in plural.
Line 19: Add "by" in front of "using".
Line 20: This is abstract. It should be informative for readers apart from the rest of the Manuscript. Suggest to specify here which chemical compounds are examined in current research.
Line 21: Suggest to replace "conducted" with "also monitored". It seems more appropriate to me.
Lines 21-24: Unify used style i.e. all names of fatty acids should be given with the first capital letter or not.
Line 25: You already listed SFAs in Lines above so they can not be "minor components" as you said here, right? In addition, later in the text it will be revealed that you had more than 60% of SFAs among alll UFAs so this statement is completely incorrect. Suggest to delete it or to specify which FAs were minor components in examined propolis.
Line 27: I think it can not be just "luteolin-7-glucoside" but "luteolin-7-O-glucoside"?? Check/correct.
Line 28: Add ) after 0.54%.
Line 28: Suggest to replace "and the" with "wile".
Line 29: Here you said that you examined antioxidant properties of "extract" but lately we will found that you actually examined more than one extract. So, than it mast be "extracts" in plural? Check/explain/correct.
Line 29: DPPH should be given with label for radical i.e. DPPH. or as "DPPH radical". Correct and apply this through the whole Manuscript.
Line 30: As I said this Line is not in accordance with previous. So, did you examine one or more extracts?
Line 33: Give the value obtained for alpha-tocopherol here. This compound just "jumped out" here without any previous informations.
Lines 34-35: And what about pesticide residues in samples? Are they present or not? Give some additional informations here.
Line 41: As I said in my the first comment this statement about Apis mellifera should be correct but here also name should be given in Italic.
Line 43: "for" not "since".
Lines 48-51: Authors should provide adequate reference(s) for given statements here.
Line 55: typo - "flavor" without capital letter F.
Line 58: Suggest to delete "All" and to rewrite as follow: "In general, the propolis contains flavonoids and..."
Line 61: Put "p-" in Italic here.
Lines 63-67: Unclear and confusing part. I do not understand how/why is Mexican honey relevant for propolis research from Jordan?? Chek/explain/correct/rewrite. In addition, again, Latin name should be given in Italic here.
Line 70: "O" letter in name of glycosides must be write in Italic style. Correct and apply this through the whole Manuscript.
Line 76: I think that name of this subsection should be "Results and Discussion" since you do not have separate Discussion section.
Line 79: I think that something is wrong with this statement "the hive collected from the bees". Check/explain/correct.
Line 83: "Jordanian propolis".
Line 85: what is meaning of "around 19 compounds"? Did you determine 19 compounds or not? There is no "around" about that.
Line 85: Finish sentence after "identified" and start the new one as follow:" The most of compounds..."
Line 89: Something is missing here after "for the first time" some name of fatty acid. Check/correct.
Line 89: Put "cis" in Italic in chemical names. Correct and apply this through the whole Manuscript.
Line 90: As I said in abstract section this statement about SFAs as minor component in propolis is not in agreement with results that authors presented in Table. So, it must be checked and explained/corrected.
Lines 91-95: The same issue as in Lines 21-24.
Line 95: Again authors repeat statement about SFAs as minor components.
Line 96: The same as in Lines 91-95.
Lines 97-98: This this years after references. They are surplus. Erorr is repeating in Manuscript so, check and correct all.
Line 100: How/why now you are mentioned here phenolic compounds among FAs and GC analysis?
Line 100: Delete "the" in front of "10. You can not use the for plural. Also, now it is 10 compoundsa dn previously it was 19 compounds. So what is true?
Line 101: Delete "as". Also, if it is hexadecanoic acid than it is saturated FA so it can not be 9,12-hexadecanoic but 9,12-hexadecadienoic. Check/correct.
Line 102: Why/how now you are talking about "esters" here and not about particular FA? Check/explain/correct.
Line 105: Delete "they identified" and replace/rewrite as follow: "... were 14 fatty acids (GC-MS method) like ..."
Lines 108-109: Again here we have esters, why?? Linoleic acid certainly is not the ester! Check/explain/correct.
Line 110: "Oman".
Line 110: Add , after "hydroxy acids".
Line 111: Add , after "fatty acids".
Line 112: Suggest to finish sentence after "chalcones" and to start the new one as "They observed...".
Line 112: Delete "the" in front of "propolis" here.
Lines 113-114: I think that GC-MS method is more advanced compared to GC-FID. So, this is not some particular advantage of your research.
Line 118: Delete "that the".
In Table 1 put "cis" in Italic.
Line 128 and Table 2: As I said I think that for all glycosides names you have missing letter O in names. Check and correct all.
Line 143: "pesticides" in plural. Also, please do not use ppb but SI unit. Correct all in text and Table 3.
Line 148: Which guidline? Specify here.
Lines 1596-160: Confusing part of text. First authors talking about only "hexane extract" (Line 157) and than we will find out that you examine also ethanolic extracts (Line 160). In addition, why authors did not provide results for vitamin C since since they use it as positive control?
Line 164: Something is wrong with this statement "The ROS production ... Colombian propolis". It is comletely incorrect. Check/explain/correct.
Line 167: Yea, chemical composition is different but all of them contained phenolics which are the most powerful antioxidants, right?
Line 176: Put "In-vitro" in Italic here.
Line 177: "no" not "not".
Line 178: Again, confusing one "extract" or all "extracts" or some of them?? Check/correct/explain.
Line 179: "It should be "Results of xanthine oxidase activity are presented in the Table 4". Also, I suggest to authors to introduce abbreviation XO here not later in the text.
Line 181: "extracts" not "the extract".
Line 185: Define abbreviation CAPE.
Line 191: Flavonoids are phenolic compounds so it is surplus here. It can be "flavonoids and phenolic acids" if you want to mention both.
Line 200: Explain how/why 70% extract had the highest DPPH activity. Also, give some explanation for results of enzymes assays used in current study.
Line 216: "pesticide residues" not just "pesticide".
Lines 306-314: Used abbreviated names for authors here. For instance, R.R.N. for Rajashri R. Naik.
Author Response
Thank you for comments
Please refer the uploaded copy for the response

Reviewer 2 Report
It is important to add in the review a comparison of the use of propolis as a medicine in different parts of the world (at least in the countries of Europe and Asia) in both scientific and traditional medicine.
Compare obtained data on chemical composition for propolis from Jordan with data for other countries.
Explain the reasons for the ingestion of pesticides in propolis.
These data would significantly raise the level of the article and would show the breadth of knowledge about propolis as an important medicine and beekeeping product.
Author Response
Thank you for comments. The answers to comments are given below
1. It is important to add in the review a comparison of the use of propolis as a medicine in different parts of the world (at least in the countries of Europe and Asia) in both scientific and traditional medicine. --> Text is updated
2. Compare obtained data on chemical composition for propolis from Jordan with data for other countries. --> Complied
3. Explain the reasons for the ingestion of pesticides in propolis. --> It is due to the use of pesticide, we did not ingest.
These data would significantly raise the level of the article and would show the breadth of knowledge about propolis as an important medicine and beekeeping product. --->
Thank you for comments and suggestions
Reviewer 3 Report
The results presented by this study will be attracting the interest of Molecules audience. The manuscript is well written and structured, and the reading and results are fluently.
However, I would advise some necessary changes to the manuscript in order to improve its quality.
My suggestions and specific comments are included below.
Line 127-131 - Are the results consistent with other studies? References to other studies with similar results should be added.
Line 147-148 - "The amount of pesticide in the propolis is within the acceptable range as per guideline"- should be added the acceptable range according to guideline. According to which guide are the values obtained reported?
Line 206 " collected from beehives located at AlGhour regions in Jordan from March to July 2018" – should be mentioned also the regions from which the samples were collected.
Line 208 – "The ground propolis" should be adding the type of grinder used.
Line 210 - "and the solvent was evaporated using rotator evaporator". – should be added the model, manufacturer and manufacturing country of rotator evaporator
Line 216 - "The α-tocopherol content was determined in the lipid fraction"- should be added how lipid fraction was extracted. Which equipment was used?
Line 222 – "45 minute with constant shaking" – should be added the equipment used for shaking.
Line 285 - "The absorbance of the solutions was measured at 517 nm using hexane as blank" – should be added the equipment used for measurements.
Line 303-304 - The study also suggests that the pesticide content were below the permissible limit due to the controlled use of pesticide in the crops - should be more specific. Which are the permissible limits for the pesticides studied? These limits should be added in lines 147-148 in discussions part.
I appreciate the work done in elaborating this study and I recommend in the future more research on the therapeutic potential of propolis studied in this paper.
Author Response
Thank you for comments
Please find enclosed herewith answers to the reviewer

Reviewer 4 Report
Article description
Authors performer research of chemical composition and biological activity (DPPH and xanthine oxidase inhibition) of Jordanian propolis 70% and 50% ethanol in water extracts and lipids fraction. Composition was analysed by HPLC-DAD (polyphenols), LC-MS/MS (pesticides) and GC-MS (lipid fraction). In results, authors exhibited potent radical scavering and xanthine oxidase inhibition activity. Chemical composition was partially investigated according to present standards of research.
Review summary
Article is interesting for propolis researchers due to lack of significant data from investigated area, but it contains also some mistakes and misconceptions. The biggest disadvantage of the article is only one researched sample, however due to very partially data from Jordanian area this is acceptable. Moreover, chemical composition was only partially investigated according to present standards. In my opinion, article may be acceptable after correction and supplementation of lack data, especially LC-MS/MS analyse of 70% and 50% ethanol in water extracts.
Major issues
- Please add material and reagents section! Without list of used chemicals and standards article cannot be published.
- Please use italics for cis/trans isomers
- Please describe potential plant precursor of investigated propolis samples. Presence of flavonoid may suggest poplar origin while carnosic acid suggest presence of Cupressaceae family.
- Line 58-59: „All the propolis in general contains flavonoid and diterpenes, as these chemical constituents are present in the flowers and other parts of the plant”
This statesmen is should be corrected. Most of worldwide propolis is rather mix of polyphenols such as flavonoids and free cinnamics acids and their derivatives. Diterpenes rarely are dominating components of propolis, usually they are secondary or even trace components in comparison to polyphenols. However, there are known types of propolis without polyphenols and contain almost same diterpenes. Moreover, typical propolis plants precursors are physiological exudates from leaf or flower buds or pathological exudates from injured plants parts such as barks. There are known kinds of bee glue originated from herb plants exudates, but there are rather rare.
- Line 63-72: Why did you write about honey? How is connection between propolis and honey? Why did you compare Mexican and Jordanian propolis? How is general conception? It will be better compare Jordanian propolis to near area such as Egypt (especially Mediterranean area). Think, that this fragment should be rewritten.
- In the future, authors may try compare fatty acid of propolis with fatty acid of beeswax. This investigation may allow to exclude wax origin fatty acid.
- Authors should also perform LC-MS/MS analyses of samples composition. Why authors did not perform this analyse while LC-MS/MS research of pesticides was done? Rich data of MS and UV spectres of propolis flavonoid and another polyphenols usually allows to identification more components, especially main components. Beetwen chrysin and pinocembrin in chromatogram there are a lot of strong peaks. According to my own experience they may be isomers of caffeic acid prenyl and benzyl esters (poplar markers) or additional flavonoids such as sakuranetin or pinobanksin 3-O-acatate (strong poplar marker).
- In future, investigation should be perfomed with more samples.
Minor issues
- Line 78 – Nontoxic for who?
Line 41, 65 – Please use italics for Apis mellifera.
Line 87 – what is FAME? Please explain short when is used the first time in the text.
Line 110 – Please correct ‘oman’ to ‘Oman’.
Line 130 – Please correct hisperidin to hesperidin
Author Response
Thank you for comments
Please refer enclosed document
thank you

Round 2
Reviewer 1 Report
No further comments.
Reviewer 4 Report
Article description
Authors investigated fatty acid and polyphenols composition as well as pesticide residues, DPPH radical scavenging and xanthine oxidase inhibitory ion activity of Jordanian propolis.
Review summary
Some major changes was introduced, however, there are still some things to correction. All of them there are listed in the next paragraphs.
Major issues
There are still unidentified major components of extracts! Expecially peaks between pinocembrin and chrysin. In my own experience they probably should be caffeic acids esters (prenyl/isoprenyl and benzyl, UV max = ~320-324 nm and mass of deprotonated molecules 269 for benzyl ester and 247 for prenyl/isoprenyl esters) and/or sakurantin (UV = ~285-290 nm and ion mass 285). Sometimes in this area is also presented component as acacetin (ion mass = 283), pinocembrin chalcone etc. There are a lot of literature which allow to identify these components. In my opinion at least preliminary identification should be performed because it not required a lot of work today.
Line 65 – Please change “are present in the flowers and other parts of the plant” on "are present in physiological and pathological plant exudates of buds (flowers and leaf), bark and other organs".
Direct plant precursors of propolis are plant exudates not organs such as flowers.
Line 69-87 – This paragraph have to been rewritten. It is no fluid nor have logical structure.
Line 196-317 – Please explain how relative percentage of polyphenols was calculated. Why authors did not perform full quantification of known components?
This manuscript is a resubmission of an earlier submission. The following is a list of the peer review reports and author responses from that submission.